# Surface Gene Mutations of Hepatitis B Virus and Related Pathogenic Mechanisms: A Narrative Review

**DOI:** 10.3390/v17070974

**Published:** 2025-07-12

**Authors:** Tingxi Yan, Yusheng Zhang, Huifang Zhou, Ning Jiang, Xiaotong Wang, Wei Yan, Jianhua Yin

**Affiliations:** 1Department of Laboratory Medicine, Naval Medical Centre, Naval Medical University, Shanghai 200052, China; ytx1034453420@163.com (T.Y.); 16655254550@163.com (Y.Z.); 13671552109@163.com (H.Z.); jiangning990921@163.com (N.J.); wxtone@foxmail.com (X.W.); 2Department of Epidemiology, Naval Medical University, Shanghai 200433, China

**Keywords:** hepatitis B virus, gene mutation, endoplasmic reticulum stress, unfolded protein response

## Abstract

Liver cancer has high incidence and mortality rates worldwide, with hepatocellular carcinoma (HCC) being the main histological subtype, accounting for 90% of primary liver cancers. The high mutation rate of viruses combined with endoplasmic reticulum stress may lead to the occurrence of cancer. Hepatitis B virus (HBV) infection is one of the most important pathogenic factors of HCC. The carcinogenic mechanisms of HBV have been widely studied. Among these mechanisms, immune escape and vaccine escape caused by mutations in the HBV S gene have been reported in numerous studies of patients with chronic hepatitis B. In addition, pre-S1/S2 mutations and surface protein truncation mutations may activate multiple signaling pathways. This activation leads to the abnormal proliferation and differentiation of hepatocytes, thereby contributing to the development of HCC. This review aims to integrate the existing literature, summarize the common mutations in the HBV S gene region, and explore the related pathogenic mechanisms.

## 1. Introduction

Hepatitis B virus (HBV) infection is a global public health issue and can cause chronic hepatitis, liver cirrhosis, and hepatocellular carcinoma (HCC). The World Health Organization estimates that, in 2022, 254 million people were living with chronic hepatitis B, with 1.2 million new cases and approximately 1.1 million deaths reported that year [1]. Although vaccines can prevent HBV infection, chronic cases remain a major global health challenge [2].

Existing studies indicate that, under vaccine pressure, hepatitis B surface antigen (HBsAg) can mutate to evade immune clearance [3]. HBsAg is encoded by the S gene, which is the main component of the viral envelope and serves as a key target of the host’s immune response. Mutations in the S gene region (such as pre-S/S deletion, G145R, the T123N, etc.) occur frequently in patients with chronic hepatitis B, especially after vaccination or antiviral treatment [4]. Recent studies have revealed that specific mutations (such as sG145R and the deletion of pre-S2) are significantly associated with the risk of HCC. These mutations not only affect the life cycle of HBV but also may promote the occurrence of HCC by altering the host’s immune response, such as by impairing immune recognition or inducing chronic inflammation. Therefore, studying the related mechanisms of these mutations is of great significance for the development of new prevention and control strategies.

## 2. Structure and Function of HBV S Gene

### 2.1. S Gene Encoding Domain

The HBV genome is approximately 3.2 kb long and consists of partially double-stranded circular DNA with four overlapping open reading frames (ORFs): pre-S/S, preCore/Core, Pol, and X. Among these, the S gene is located in the S region of an ORF and partially overlaps with the polymerase gene (P gene) [5].

The S gene of HBV is transcriptionally regulated by three promoters, S1, S2, and the major promoter, which generate three distinct mRNAs through differential transcription initiation sites. These mRNAs encode three viral envelope proteins: large (L-HBsAg), medium (M-HBsAg), and small (S-HBsAg) surface proteins [6]. Among these proteins, S-HBsAg is the most abundantly expressed, constituting over 90% of viral envelope components. In contrast, L-HBsAg and M-HBsAg have unique N-terminal extensions that confer functional diversity, the pre-S1 and pre-S2 regions, respectively [7].

Structurally, S-HBsAg mediates the morphogenesis of both subviral particles and infectious virions (non-infectious particles devoid of viral genomes mediate immune tolerance and represent targets for diagnostic and therapeutic interventions). In contrast, M-HBsAg contains an additional glycosylated pre-S2 domain, and L-HBsAg incorporates both the pre-S1 and pre-S2 domains appended to the M-HBsAg framework [8]. These domains play distinct roles in the HBV life cycle: the pre-S1 and pre-S2 regions of L-HBsAg are critical for viral attachment and host cell entry, while M-HBsAg and S-HBsAg primarily facilitate viral assembly and secretion [9].

Notably, genetic variations in HBsAg (e.g., subtype-specific polymorphisms or mutations) can alter its antigenic properties and immune evasion capacity, thereby modulating HBV infectivity and transmission dynamics [9].

During HBV infection, mutations in the S gene can alter the antigenicity of HBsAg through mechanisms such as the introduction of additional glycosylation sites or variations in antigenic epitopes, thereby impairing the recognition and clearance of the virus via the host immune system. For instance, specific mutations may increase the number of glycosylation sites on HBsAg, which interferes with antibody binding and facilitates immune escape [10]. Additionally, HBsAg domains are strongly linked to HBV-associated carcinogenicity. In HCC, S gene mutations are considered a critical oncogenic mechanism.

### 2.2. HBsAg Plays a Key Role in the Virus Life Cycle

HBsAg plays a pivotal role in the HBV life cycle, particularly in viral attachment, immune evasion, and virion assembly. Specifically, HBsAg facilitates viral attachment and entry into host cells by interacting with surface-specific receptors. For instance, the pre-S1 domain of HBsAg binds to the sodium taurocholate co-transporting polypeptide (NTCP) on hepatocytes, which is a critical receptor-mediated step for HBV entry into target cells [11] (as shown in Figure 1).

Second, HBsAg also plays an important role in immune escape. Hepatitis B surface protein synthesis occurs in the endoplasmic reticulum (ER), in which the amount of protein far exceeds that of virus particles, and high concentrations of HBsAg can suppress the host immune response by forming immune complexes (HBsAg-CICs), thus resulting in persistent HBV infection [12]. It has been found that certain mutations in HBsAg can alter its antigenicity, which results in the antibodies produced by the host being unable to effectively recognize and neutralize HBsAg [13]. In addition, glycosylation changes in HBsAg are also thought to be an important mechanism leading to immune escape, which makes HBsAg difficult to detect in blood and further enhances HBV viability [14].

Finally, HBsAg is also crucial in the assembly and release of viral particles. HBsAg forms virus-like particles (VLPs) via self-assembly. These particles are not only the main antigen of HBV but also an important basis for vaccine development. Studies have shown that HBsAg assembly is regulated by a variety of factors, including endogenous signaling pathways in the host cell and the genome structure of HBV itself [15]. By regulating HBsAg expression and assembly, the immunogenicity of the vaccine can be improved, thus enhancing the protective effect against HBV.

In conclusion, HBsAg plays multiple central roles in the life cycle of HBV, and changes and mutations in its domain not only affect the infectious ability of the virus but are also closely related to the host immune response. Furthermore, hepatitis B cure paradigms comprise two distinct endpoints: functional cure and complete cure. Functional cure is defined as sustained undetectable serum HBV DNA and HBsAg, with normal alanine aminotransferase (ALT) levels persisting for ≥24 weeks after treatment cessation. Complete cure signifies the total eradication of covalently closed circular DNA (cccDNA) and integrated HBV genomes from all host tissues. Achieving functional cure represents a primary therapeutic goal for chronic hepatitis B. Future research should elucidate HBsAg’s biological functions to harness its potential in developing novel therapeutic vaccines, which is a strategy that may ultimately achieve either a functional or complete cure.

## 3. Epidemiological Characteristics of Mutations in Region S

### 3.1. Pre-S/S Gene

The pre-S/S gene contains three in-frame initiation codons. These codons encode the S-HBsAg, the M-HBsAg containing the pre-S2 sequence, and the L-HBsAg containing both the pre-S1 and pre-S2 sequences. The L protein is translated from a 2.4 kb pre-S1 RNA transcript, while the M and S proteins are translated from a slightly shorter 2.1 kb S RNA transcript. The S protein consists of 226 amino acids (aa), while the M protein has an additional 55 aa (pre-S2 region). The L protein is an extension of the M protein, with it having an additional 108–119 aa depending on the genotype (i.e., pre-S1 region) [13] (Figure 2).

### 3.2. Mutation Mechanisms of Hepatitis B Surface Antigen S Gene

HBV has a unique biological process characterized by error-prone reverse transcriptase (RT) activity and an abnormally high daily production of viral particles. HBV reverse-transcribes an RNA intermediate to synthesize its DNA genome, which is a distinctive feature among DNA viruses. Its DNA replication error rate is much higher than that of other DNA viruses because the RT lacks a proofreading function. However, the nucleotide substitution rate is lower than that of other RNA viruses due to the overlapping nature of the genome. The estimated nucleotide substitution rate is approximately 1.4–3.2 × 10^5^ per locus per year [16]. As HBV mutation progresses, escape-related mutations are found continuously with immune stress and/or antiviral drug use, and antiviral drug resistance thus becomes a major factor limiting drug efficacy. In addition, due to the overlapping of viral polymerase and envelope reading frames in the HBV DNA genome, selected nucleos(t)ide analogue (NA) resistance-associated mutations in the polymerase catalytic domain often result in important changes in the neutralizing antibody binding domain of HBsAg [3].

## 4. Common Mutation Types

### 4.1. S Gene Mutation

Hepatitis B can cause chronic infections. It can be transmitted through contact with infected body fluids, and it can also be passed from mother to baby. All infants should be vaccinated against hepatitis B as soon as possible after birth (within 24 h), which should be followed by two or three doses of hepatitis B vaccine at least 4 weeks apart [17].

The hepatitis B vaccine can effectively prevent HBV infection. With the widespread use of the vaccine and the continuous evolution of the virus, Italian researchers in 1988 first associated the phenomenon of “vaccine escape” with a point mutation, i.e., glycine replaced by an arginine residue at position 145 (G145R) [18]. Since then, an increasing number of escape sites have been confirmed. G145R is located in the “a” determining region of HBsAg, which consists of aa 124 to 147 and is located in the central core of SHB (aa 99–169, also known as the major hydrophilic region, MHR) [19].

The MHR has band T-cell-associated cellular epitopes and point mutations, substitutions, or deletions in the S gene region. The MHR can also cause immune escape or diagnostic escape. One important finding based on global HBsAg sequencing methods is the identification of many novel MHR mutations. A total of 345 different aa changes were identified within the MHR [20]. All 51 known “a” determinant mutations associated with diagnostic failure and immune escape are included in a library, with 62 of the 345 aa substitutions identified representing previously unrecognized MHR mutations [21]. It has been reported that escape mutations of HBV are mainly found in the first loop of the MHR, while mutations induced by immune stress are in the second loop [21]. The mutation efficiency of the two loops of the MHR is different, which may be due to the shift in genotype that occurred when the relevant case study was conducted. MHR mutations have been reported to be associated with the genotype, and a study of 4.4 million Dutch blood donors identified HBsAg-negative but HBV DNA-positive individuals at various stages of HBV infection. The authors reported multiple S gene escape mutations in these subjects, especially genotype D-positive components [22]. Other studies have shown that the prevalence of MHR mutations varies with subgenotype. The prevalence of MHR mutations (including escape mutations and overlapping RT mutations) in the virus surface antigen of the C5 subgenotype was significantly higher than that of other subgenotypes. The C5 subgenotype is common in Guangxi, China [23].

### 4.2. Pre-S1/S2 Deletion Mutation

Among the three surface proteins, the pre-S2 structure of M-HBsAg is closely linked to virus particle production. The pre-S1 structure of L-HBsAg plays a key role in virus particle assembly and secretion and mediates virus entry into cells. Some naturally occurring deletion mutations occur in the pre-S1 gene segment (ranging from nucleotides (nt) 2854 to 3210) or the pre-S2 gene segment (ranging from nt 3211 to 3221 and nt 1 to 154), and these proteins contain deleted pre-S1 domains (referred to as pre-S1 mutants) or deleted pre-S2 domains (referred to as pre-S2 mutants) [24].

Previous studies demonstrated that ground glass hepatocyte (GGH) types I and II are correlated with distinct HBV surface antigen expression patterns in chronically infected cells. Type I GGH is linked to partial pre-S1 deletions, while type II GGH is associated with partial pre-S2 deletions [25]. Type I GGHs are typically scattered individually in hepatic lobules and express surface proteins in an “inclusion body-like” pattern. In contrast, type II GGHs display surface proteins at the cell border with a distinct expression pattern. They are consistently found in clustered nodules and are often associated with cirrhosis or HCC [26]. This suggests that type II GGHs are involved in clonal proliferation and carcinogenesis. A later study showed that GGHs carry pre-S deletion variants that accumulate in the ER and induce ER stress signaling [27].

The aa sequence between positions 3 and 77 in the pre-S1 region is involved in virion secretion during infection. When a pre-S1 defect overlaps this region, it causes a mild defect in HBsAg secretion. The pre-S region also includes the initiation site of the S region; thus, deletions may affect the synthesis of other surface proteins [27]. Consequently, pre-S deletion mutants reduce the synthesis of S-HBsAg and M-HBsAg. Since the secretion of L-HBsAg depends on the presence of excess S-HBsAg and M-HBsAg, deletions in the pre-S region lead to the accumulation of L-HBsAg in the ER [28]. This accumulation causes ER stress and triggers unfolded protein responses (UPRs) that damage hepatocytes. Prolonged damage may ultimately result in HCC.

### 4.3. Truncation Mutations

Because the S gene region and P-ORF overlap completely, when a P gene sequence mutation occurs, it leads to corresponding S gene region changes [29]. Some mutations in the P gene region encode a stop codon in the corresponding S gene region, thus resulting in the non-expression of the C-terminal region of the HBV surface protein, and these mutations are known as truncation mutations. rtA181T/sW172, rtM204I/sW196, and rtV191I/sW182 are the most frequently reported resistance mutations with C-terminal truncation, which have carcinogenic potential [30]. The rtA181T/sW172 mutant is a point mutation (TGGCTC→TGACTC) at position 181 of the P gene, which causes an alanine (A)→threonine (T) mutation in the polymerase. Due to the complete overlap of the S gene and RT domains, the mutation (sW172) replaces aa tryptophan (W) at position 172 of the S gene with a stop codon (UGA), resulting in the truncation of the last 55 aa of the S protein [31].

C-terminal truncated medium size surface protein (MHBst) and L-HBsAg have been reported as transcriptional activators leading to hepatocyte proliferation [32,33,34]. The results of transgenic mouse studies and hepatoma cell cultures indicate that the MHBst protein retained in the ER can trigger the PKC-dependent activation of the c-Raf-1/Erk2 signaling cascade. This leads to the induction of activator protein 1 (AP-1) and nuclear factor kappa B (NF-κB) transcription factors and enhances the proliferative activity of hepatocytes [35,36]. Table 1 summarizes the common types of surface protein gene mutations, their mechanisms, and their clinical significance.

## 5. Hepatitis B Virus S Gene Mutation and Pathogenesis

### Hepatitis B Surface Antigen S Gene Mutation Leads to Virus Replication and Genome Integration

HBV–DNA integration is an important part of HBV carcinogenesis. Studies have shown that HBV S gene mutations are closely related to HBV–DNA integration in the host genome, which may lead to genomic instability in liver cells, which, in turn, leads to carcinogenesis [37]. For example, HBV integration sites often occur near tumor driver genes, such as *TERT* and *TP53*, which may lead to the abnormal expression of these genes, thereby promoting the malignant transformation of hepatocytes [38].

## 6. Occult Hepatitis and Chronic Inflammation

Occult HBV infection (OBI) is defined as the absence or undetectable presence of HBV DNA in the serum but persistent presence in liver tissue in HBsAg-negative individuals [39]. S/pre-S gene mutation is a common cause of OBI. Different mechanisms of HBV mutation may explain the OBI presentation in these patients: (1) the large deletion of pre-S1 reduces viral replication and expression; (2) S gene mutation reduces HBsAg antigenicity; and (3) mutant HBsAg may elicit low affinity or non-neutralizing antibodies, thus interfering with HBsAg reagent detection [40].

It has been reported that the prevalence of OBI among the general population in Guangdong, Hainan, and Long’an County of China is 2.0% (6/294), 3.4% (68/1995), and 11.5% (6/52), respectively [41], and the prevalence of OBI among Asian blood donors is 1:570–1:7517 in China [42]. OBI detection is useful as it can reduce the transmission of HBV. In addition, OBI has lower expression levels of HBV DNA in the serum, but there is still unsecreted DNA in the liver, which may lead to HBV recurrence. HBV reactivation diagnosed as OBI usually requires early treatment. Another study [43] found that patients with OBI/Hepatitis C Virus (HCV) co-infection had significantly more fibrosis and necrotic inflammation than those with HCV infection alone, regardless of whether they received interferon treatment or not, and their risk of HCC was also significantly higher.

The progression of hepatitis B is characterized by distinct acute and chronic phases. During acute infection, the host immune response—particularly its viral clearance capacity—may eliminate the virus. In contrast, chronic infection develops through viral persistence mechanisms, including (i) the reduced exposure of viral antigens below immunological detection threshold and (ii) immune-evading mutations in HBV epitopes. These mechanisms collectively drive sustained hepatic inflammation [44]. The cytokines produced via inflammation may contribute to HCC. Tumor necrosis factor α (TNF-α) is one of the most characteristic tumorigenic cytokines in hepatocarcinogenesis. It activates the NF-κB signaling pathway. NF-κB has dual functions in hepatocarcinogenesis. NF-κ b-induced inflammation is known to promote tumorigenesis. I-κB kinase (IKK) beta knockout mice, with reduced activation levels of NF-κB in hepatocytes, exhibit enhanced hepatocyte apoptosis and compensatory proliferation, making them susceptible to HCC [45]. These mice are deficient in essential NF-κB modulators, lack hepatocyte NF-κB activation abilities, and can regulate apoptotic enzymes and apoptosis-related proteins, which reduces HCC development, thus suggesting that the anti-apoptotic function of NF-κB may prevent HCC development by inhibiting compensatory proliferation [46,47]. Persistent inflammation during OBI, the risk of HBV reactivation, and co-infection with other hepatotropic viruses collectively contribute to HCC pathogenesis.

## 7. ER Stress and HCC

The ER accounts for 60% of the cell membrane and is the intracellular organelle responsible for protein synthesis, folding, transport, and maturation. It establishes a tightly linked network with other intracellular organelles, such as mitochondria, the Golgi apparatus, endosomes, peroxisomes, and plasma membranes [48,49]. In terms of functions of the ER, the correct folding and modification of proteins are its most important roles. Only correctly processed proteins are exported to the Golgi apparatus, and misfolded proteins are retained in the ER for further processing or degradation [50,51].

During chronic hepatitis B infection, with a pre-S gene mutation, the synthesis and secretion of S-HBsAg and M-HBsAg are reduced, and the synthesis of L-HBsAg is increased, which cannot be secreted when expressed alone but remains in the cell as luminal particles [52,53]. Under normal circumstances, the ER has an equilibrium between its ability to fold and unfold proteins [54]. Unfolded or misfolded proteins accumulate and aggregate in the ER, leading to ER stress [55]. The UPR is an adaptive mechanism of the ER that relieves ER stress and maintains its homeostasis [56,57]. Excessive UPR and ER stress can lead to liver inflammation, cell death, tissue damage, and fibrosis, thus resulting in the development of various liver diseases [58]. In addition, there is growing evidence that ER stress-mediated liver inflammation can promote the development of liver steatosis and injury [59].

ER stress-related proteins inositol demand protein 1α (IRE1α), activating transcription factor 6 (ATF6), PRKR-like endoplasmic reticulum kinase (PERK), eukaryotic initiation factor 2 (eIF2), X-Box binding protein 1 (XBP1), C/EBP homologous protein (CHOP), and glucose-regulated protein 78 (GRP78) were significantly upregulated in hepatitis B surface mutants. ER stress causes calcium to be released from the ER into the cytoplasm, thereby inducing a significant decrease in the expression of antioxidant proteins, such as intracellular reactive oxygen species (ROS), superoxide dismutase (MnSOD and CuZnSOD), heme Oxygenase-1 (HO-1), and catalase [60].

There are three UPR transducers in the ER: PERK, IRE1, and ATF6 [61] (as shown in Figure 3).

PERK is a transmembrane protein located in the ER. Under normal conditions, PERK binds to ER partner glucose regulatory protein GRP78/binding immunoglobulin protein (BiP) [62]. When ER stress occurs, BiP dissociates from PERK, which changes from a monomer to an oligomer state when activated [63]. Once activated, PERK phosphorylates eIF2α, which is associated with reduced translation/protein synthesis. Upon the phosphorylation of eIF2α, activated transcription factor 4 (ATF4) mRNA is translated and induces UPR target genes associated with apoptosis [64].

IRE1 is a type I transmembrane protein with dual kinase and endonuclease functions [65,66]. Under stress conditions, GRP78 is sequestered on misfolded or unfolded proteins in the ER and then releases IRE1. Its endoribonuclease (cytoplasmic RNAse domain) produces a transcriptional activator called XBP1 to increase protein folding ability or lead to the transcriptional induction of genes encoding protein-degrading enzymes [67]. At the same time, IRE1 kinase activity induces apoptosis signal transduction kinase-1 (ASK-1), Jun-N terminal kinase (JNK), and p38 mitogen-activated protein kinase (p38 MAPK), thus leading to apoptosis [68,69].

ATF6 is a type II transmembrane protein with the N-terminal located in the cytoplasm [70,71]. ATF6 is covered by GRP78, like PERK and IRE1a, but under ER stress, ATF6 is released and transferred to the Golgi apparatus. After cleavage via S1P and S2P in the Golgi apparatus, the functional parts of ATF6 are transported to the cytosol and then the nucleus [72]. In the nucleus, ATF6 induces ER-associated degradation and chaperone proteins [73].

NF-κB has been proposed to play a major role in the progression of inflammation-related cancers [74]. The activation of NF-κB is not only involved in the traditional degradation of IκB but is also regulated by p38 mitogen-activated protein kinase (MAPK) [75]. The inhibition of p38 MAPK does not affect the nuclear translocation of NF-κB but inhibits NF-κB DNA binding activity and attenuates cyclooxygenase-2 induction [76].

The Wnt/β-catenin cascade is one of the major signal transduction pathways regulating liver homeostasis, regeneration, and tumorigenesis [77]. In short, in the absence of the Wnt ligand, most cells contain β-catenin sequestered outside the cell membrane. Cytoplasmic β-catenin forms complexes with axonectin, tyrosine kinase 1, and glycogen synthesis kinase 3, which ubiquitinate β-catenin and degrade it via proteasomes [78]. During ER stress, Wnt ligands bind to frizzled receptors and interact with lipoprotein receptor-related proteins 5/6. This interaction blocks the phosphorylation of the β-catenin destruction complex, thus disrupting its function. Consequently, β-catenin accumulates in the cytoplasm and translocates to the nucleus, where it forms a complex with T-cell factor/lymphoid enhancer factor (TCF/LEF) transcription factors. The activated β-catenin/TCF/LEF complex then induces the transcription of Wnt target genes, thereby triggering downstream cellular responses [79].

## 8. Discussion

Mutations in the HBV surface proteins significantly alter the virus’s immune epitopes, thus facilitating widespread immune escape and presenting significant clinical challenges. These challenges include a higher rate of diagnostic and treatment failures in chronic hepatitis B due to surface protein mutations. Sequencing studies across diverse ethnic groups reveal a high mutation rate within the MHR [21], with participants often harboring multiple mutant strains. This elevated mutation rate may also reduce vaccine efficacy, thus leading to the persistence and progression of chronic hepatitis B and an increased risk of HCC. To address this, multi-parameter detection strategies are recommended for chronic HBV infection. Studies have identified over three linear epitopes in HBsAg, with the second loop of the “a” determinant serving as the immunodominant site [80]. Subdominant epitopes (e.g., regions outside the “a” determinant’s second loop) retain antigenicity and enhance weak anti-HBs responses in murine DNA immunization models [81]; this evidence suggests the potential of subdominant epitope-based vaccines to counteract immune escape variants.

Furthermore, ER stress plays an important role in HBV infection. When S gene mutation leads to a decrease in L-HBsAg secretion and accumulation in the ER, ER stress and UPR lead to a series of related reactions, thus resulting in HBV genome damage, cell proliferation, and apoptosis. Additionally, the intrinsically high mutation rate of the HBV genome may promote malignant transformation under certain conditions. Whether there are good indicators that help identify the degree of ER stress or better tests that indicate DNA levels in the liver remains to be studied.

Some articles suggest that S gene mutations do not reduce DNA replication levels in the liver [12,26] and that hepatitis B may reactivate when the host immunity declines. With respect to HBV itself or liver cells in recognition of the immune environment, DNA’s serological release mechanism remains to be explored. The primary management strategy for chronic HBV infection involves long-term nucleos(t)ide analog therapy, which suppresses viral replication through the competitive inhibition of the viral polymerase. Although S gene mutations, including truncation mutations induced by treatment pressure, are theoretically associated with HCC, current clinical evidence remains inconclusive regarding their direct role in accelerating HCC development [31].

Although several studies have demonstrated an association between mutations and disease progression, most rely on in vitro transfection or murine models and lack large-scale longitudinal clinical data [21,26,27]. Future research should combine multi-center clinical cohorts, in vivo models, and multi-omics methods to further reveal the key role of S gene mutations in the viral life cycle and host immune regulation while simultaneously facilitating the development of the next generation of vaccines or immune interventions targeting mutant HBsAg. Collectively, these advances will establish a theoretical foundation for precision medicine approaches to controlling chronic hepatitis B and HBV-related HCC.

This review synthesizes mechanisms by which hepatitis B surface gene mutations drive immune escape and LHB trafficking defects, which lead to ER stress and UPR response, thus resulting in liver disease. HCC occurs under the regulation mechanism of hepatocyte proliferation and apoptosis. However, it is unknown whether there is a regulatory factor irreversibly consumed in regulating the cell cycle that leads to the inactivation of tumor suppressor genes.

## 9. Conclusions

This review summarizes the common mutation types of HBV surface proteins and their pathogenic mechanisms, highlighting their roles in immune escape, vaccine escape, and HCC. HBV mutations are the result of natural selection and add complexity to disease management. Although this virus is highly variable, there may still be functionally conserved regions that are expected to serve as potential targets for future diagnosis and treatment. An in-depth understanding of these “relatively unchanged” factors will provide new insights for the precise prevention, control, and functional cure of HBV.

## Figures and Tables

**Figure 1 viruses-17-00974-f001:**
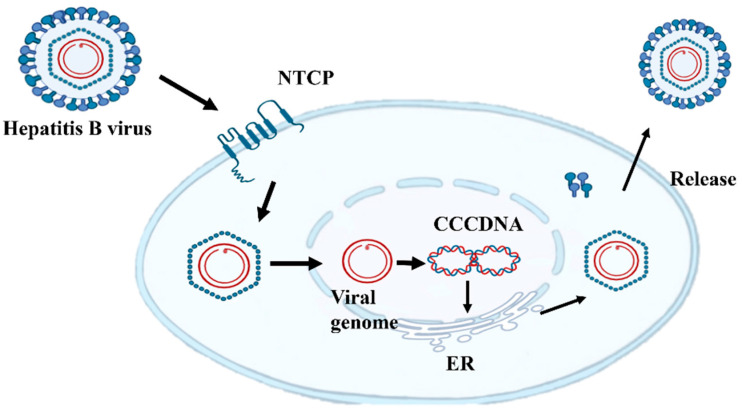
Replication cycle of hepatitis B virus (HBV), including integration into the host genome. HBV virions attach to the cell surface, allowing for clathrin-mediated endocytosis via sodium cholic acid co-transporter polypeptide (NTCP). The relaxed circular (rc) DNA genome is repaired and integrated into covalently closed circular DNA, which serves as the transcription template for transcription translation and then protein assembly into complete HBV in the endoplasmic reticulum (ER).

**Figure 2 viruses-17-00974-f002:**
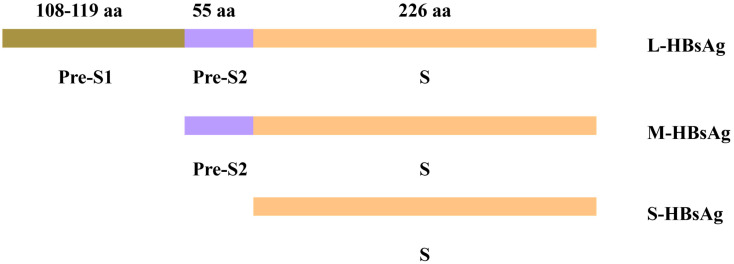
Three hepatitis B surface proteins and their corresponding gene sequences. S protein consists of 226 amino acids (aa). M protein has an additional 55 aa (pre-S2 region). L protein is an extension of the M protein, adding an additional 108–119 aa depending on the genotype (i.e., pre-S1 region).

**Figure 3 viruses-17-00974-f003:**
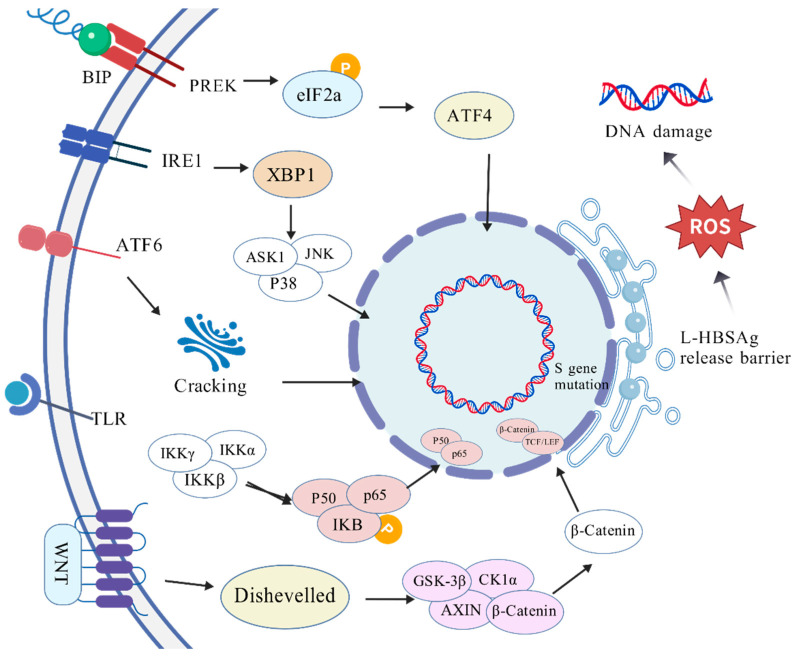
Mutations in the S gene region lead to the accumulation of large hepatitis B surface antigen (L-HBSAg) in the ER, thereby inducing ER stress and the unfolded protein response.

**Table 1 viruses-17-00974-t001:** Types of HBV surface protein mutations and their clinical significance.

Mutation Type	Representative Mutations	Molecular Mechanism	Clinical Significance
Point Mutations	G145R (“a” determinant) 345+ MHR variants	Conformational change in antigenic epitopes Disruption of B/T-cell epitopes Steric hindrance blocking antibody binding	Vaccination breakthrough Risk of occult infection Blood transfusion safety threat
Pre-S1 Deletions	nt2854-3210 deletion	Loss of pre-S1 domain (aa 3–77) Impaired virion assembly/secretion Reduced SHBs/MHBs synthesis	Type I GGH (scattered distribution) Fibrosis risk
Pre-S2 Deletions	nt3211-3221/1-154 deletion	Pre-S2 domain truncation MHBs functional loss LHBs ER retention	Type II GGH (nodular clusters) HCC risk
Truncation Mutations	rtA181T/sW172 rtM204I/sW196 rtV191I/sW182	P gene stop-codon mutations Drug-resistant mutation	HCC incidence

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
