# Peer review of "Surface Gene Mutations of Hepatitis B Virus and Related Pathogenic Mechanisms: A Narrative Review"

_viruses, 2025, doi:10.3390/v17070974_

Round 1
Reviewer 1 Report
Comments and Suggestions for Authors
Comments for authors have been added in the attached pdf.
Thanks

Significant improvement in English language is required.
Author Response
Comment 1: Title should be revised to 'Surface Gene Mutations of Hepatitis B Virus and Related Pathogenic Mechanisms: A Narrative Review' include "Hepatitis B Virus.
Response:
Thank you for this comment. According to this comment, the paper title has been revised to 'Surface Gene Mutations of Hepatitis B Virus and Related Pathogenic Mechanisms: A Narrative Review'.
Comment 2: Start of the abstract is abrupt and could be improved.
Response:
Thank you for this comment. We have revised the abstract as suggested.
Comment 3: Discussion clearly seems inadequate to address the gravity of the issue of chronic HBV infection in the light of S gene mutations
Response:
Thank you for this professional comment. We added new literatures to describe the association between S gene mutation rates with immune escape, vaccine escape, and increased HCC risk in the manuscript of Page 8, line 332 to line 337.
Comment 4: I would like to know how did you created these Figures for the article? Are these adapted, designed, or AI generated?
Response:
Thank you for this professional comment. Figure 1 depicted the structure of the HBV genome and was adapted from published literature. To avoid potential copyright issues, we have removed Figure 1 and instead described the HBV genomic structure in the text. Figures 2–4 were originally designed by our team and created using Microsoft Office PowerPoint and Generic Diagramming Platform and graphic software.
Comment 5: All references should be in uniform format.
Response:
Thank you. We have thoroughly proofread and revised all references to ensure strict formatting uniformity throughout the manuscript.
Reviewer 2 Report
Comments and Suggestions for Authors
The paper reviews the literature to synthesize common mutations in the HBV S gene region. I agreed to review this paper based on the title and a quick scan and while I have worked on hepadnaviruses in animals, I am unfamiliar with most hepatitis B research. My work is on viral evolution and viral mechanisms of cross species transmissions. Thus, I provide this review as someone who would have greatly benefited from it when I first began work on hepadnaviruses, but not someone very familiar with the material being reviewed.
I found the manuscript to be well written and clear. It provides an overview of the work that has been completed on S region mutations and how they relate to pathogenicity. I find the work to be well done and relevant.
My major suggestion is all of the acronyms is this manuscript I found overwhelming. The general ones are actually well defined (e.g., ER) but the field specific ones often are not. I think a box defining the acronyms, including major known mutations would be very helpful. Particularly because I see this work being one that primary investigators hand to new people entering their lab. A new graduate student or technician may not yet be familiar with the field specific acronyms and clear definitions in a single location would be very valuable.
In addition, I provide some minor suggestions and note some type-o’s.
Minor comments.
Initial introduction has formatting issues with the citations, some of which persist throughout.
Line 35: HBsAg is not defined
Line 90: Missing space between sentences.
Line 115-116: It could be worth mentioning the reverse transcriptase is unique given it is a DNA virus.
Line 143: “he library” to “The library”? This error occurs several places and may be a result of the citation format issue.
Line 233: Unclear what is meant by “autoimmune function”
Line 313: I suggest deleting “First of all” start with the strong statement – “Mutations in hepatatitus B virus”…
Line 342-345: Please edit this paragraph for clarity – what is meant by associated viruses? The entire paragraph also seems out of place and may be best in the conclusion?
Author Response
Comment 1: Line 35: HBsAg is not defined
Response:
Thank you for highlighting this oversight. We have now defined HBsAg (Hepatitis B surface antigen) at its first occurrence in the text. The list of abbreviations has been added to the manuscript.
Comment 2: Line 90: Missing space between sentences.
Response:
Thank you. We have implemented these modifications and conducted a comprehensive review of the entire manuscript to ensure accuracy and consistency throughout.
Comment 3: Line 115-116: It could be worth mentioning the reverse transcriptase is unique given it is a DNA virus.
Response:
Thank you for this critical observation. We have revised the sentence to accurately reflect HBV's unique replication mechanism: 'HBV utilizes reverse transcription of an RNA intermediate to synthesize its DNA genome, a distinctive feature among DNA viruses.' This modification appears in Line 122-124.
Comment 4: Line 143: “he library” to “The library”? This error occurs several places and may be a result of the citation format issue.
Response:
Thank you. We have implemented these modifications and conducted a comprehensive review of the entire manuscript to ensure accuracy and consistency throughout.
Comment 5: Line 233: Unclear what is meant by “autoimmune function”
Response:
Thank you. We have revised the sentence as follows: 'The progression of hepatitis B is characterized by distinct acute and chronic phases. During acute infection, the host immune response – particularly its viral clearance capacity- may eliminate the virus. In contrast, chronic infection develops through viral persistence mechanisms, including: (i) reduced exposure of viral antigens below immunological detection thresholds, and (ii) immune-evading mutations in HBV epitopes. These mechanisms collectively drive sustained hepatic inflammation”, specific details could be seen on line 251 in reference [44]..
Comment 6: Line 313: I suggest deleting “First of all” start with the strong statement – “Mutations in hepatatitus B virus”
Response:
Thank you. We have implemented this revision as suggested.
Comment 7: Line 342-345: Please edit this paragraph for clarity – what is meant by associated viruses? The entire paragraph also seems out of place and may be best in the conclusion?
Response:
Thank you. We have implemented this revision as suggested. The line 342 to line 345 in the manuscript has been removed.
Reviewer 3 Report
Comments and Suggestions for Authors
The authors propose a review on HBV mutation of the S protein. Some concerns should be addressed before acceptance of this manuscript.
- Line 60: The authors should briefly mention what is a subviral particle for HBV, and the relevance of this unique mechanism, also involved in immune evasion, although some information is provided in lines 84-92.
- Please correct the reference citing in the text: for ex. line 52, line 83, line 88, line 92 and so on.
- Line 118: the error rate of the HBV polymerase is higher than for other DNA viruses, but the substitution rate is however lower than for other RNA viruses, because of the overlapping nature of the genome.
- Respect to the vaccine escape mutants, it would be advisable to mention that, from an epidemiological point of view, they have not represented a threat to the effectiveness of the vaccine.
- Respect to HBsAg production through HBV integration, it would be interesting to include a brief mention on the relevance of HBsAg for detection of functional cure.
- The authors might mention that OBI can also lead to HCC.
- The Figures are not cited in the text.
- No references are included in the figures.
- Line 374: vaccine failure?
- Please correct the name of the corresponding author Jianhua Yin in Contributions, and the style.
- The references need also to be edited in the format of the journal.
Author Response
Comment 1: Line 60: The authors should briefly mention what is a subviral particle for HBV, and the relevance of this unique mechanism, also involved in immune evasion, although some information is provided in lines 84-92.
Response:
Thank you. We have added an introduction to subviral particles on line 60-61 as follows:non-infectious particles devoid of viral genomes mediate immune tolerance and represent targets for diagnostic and therapeutic interventions.
Comment 2: Please correct the reference citing in the text: for ex. line 52, line 83, line 88, line 92 and so on.
Response:
Thank you. We have implemented this revision as suggested.
Comment 3: Line 118: the error rate of the HBV polymerase is higher than for other DNA viruses, but the substitution rate is however lower than for other RNA viruses, because of the overlapping nature of the genome.
Response:
Thank you. We have revised the sentence as follows: “Its DNA replication error rate is much higher than that of other DNA viruses because the RT lacks a proofreading function. However, the nucleotide substitution rate is lower than that of other RNA viruses due to the overlapping nature of the genome.” on lines 125-127.
Comment 4: Respect to the vaccine escape mutants, it would be advisable to mention that, from an epidemiological point of view, they have not represented a threat to the effectiveness of the vaccine.
Response:
Thank you for highlighting this. We fully agree that vaccine escape variants pose no substantial threat to vaccine effectiveness from an epidemiological perspective.
Comment 5: Respect to HBsAg production through HBV integration, it would be interesting to include a brief mention on the relevance of HBsAg for detection of functional cure.
Response:
Thank you. We added the relevance of HBsAg in the detection of functional cure on lines 102-110.
Comment 6: The authors might mention that OBI can also lead to HCC.
Response:
Thank you. We have implemented this revision as suggested. We added the sentence “Persistent inflammation during OBI, the risk of HBV reactivation, and coinfection with other hepatotropic viruses collectively contribute to HCC pathogenesis”on lines 261-263.
Comment 7: The Figures are not cited in the text.
Response:
Thank you. We have implemented this revision as suggested.
Comment 8: No references are included in the figures.
Response:
Thank you. We have implemented this revision as suggested.
Comment 9: Line 374: vaccine failure?
Response:
Thank you. We have modified “vaccine failure” into “vaccine escape”.
Comment 10: Please correct the name of the corresponding author Jianhua Yin in Contributions, and the style.
Response:
Thank you. We have implemented this revision as suggested.
.
Comment 11: The references need also to be edited in the format of the journal.
Response:
Thank you. We have implemented this revision as suggested.
Reviewer 4 Report
Comments and Suggestions for Authors
Hepatitis B-related hepatocellular carcinoma (HCC) represents a significant global health burden. This review systematically discusses immune escape mutations, provides an in-depth analysis of the carcinogenic mechanisms associated with HBV surface protein mutations, and summarizes the link between the accumulation of HBV surface proteins in the endoplasmic reticulum, activation of the ER stress pathway, and HCC development.
Comments:
- Line 126 / Section 4: Common Mutation Types: For the section on "Surface Gene of Hepatitis B," it is recommended to summarize the clinical mutation types and their associated clinical significance (e.g., occurrence rate in HCC patients) in a table format.
- The discussion does not address whether nucleos(t)ide analogue therapy (e.g., Lamivudine)-induced S gene mutations are associated with the acceleration of HCC.
- Figure 1 There is number ‘5’ in Figure1. Please check it.
Author Response
Comment 1: Line 126 / Section 4: Common Mutation Types: For the section on "Surface Gene of Hepatitis B," it is recommended to summarize the clinical mutation types and their associated clinical significance (e.g., occurrence rate in HCC patients) in a table format.
Response:
Thank you. We have implemented this revision as suggested. The details of the clinical mutation types and their associated clinical significance are shown in Table 1.
Comment 2: The discussion does not address whether nucleos(t)ide analogue therapy (e.g., Lamivudine)-induced S gene mutations are associated with the acceleration of HCC.
Response:
Thank you. We have implemented this revision as suggested.
We have revised the sentence as follows: “The primary management strategy for chronic HBV infection involves long-term nucleos(t)ide analogue (e.g., Lamivudine) therapy, which suppresses viral replication through competitive inhibition of the viral polymerase. Although S gene mutations, including truncation mutations induced by treatment pressure, are theoretically associated with hepatocellular carcinoma (HCC), current clinical evidence remains inconclusive regarding their direct role in accelerating HCC development” on lines 356-361.
Comment 3: Figure 1 There is number ‘5’ in Figure1. Please check it.
Response:
Thank you. Figure 1 depicted the structure of the HBV genome and was adapted from published literature. To avoid potential copyright issues, we have removed Figure 1 and instead described the HBV genomic structure in the text.
Round 2
Reviewer 1 Report
Comments and Suggestions for Authors
Thank you for the revised manuscript.
1) The authors seem to have adequately addressed the comments.
2) However, when you say several studies, you should add references as well to support those claims. This issue was prominent in Discussion section (Paragraph 4).
3) English could be improved. There are several mistakes, for example, in last line of the abstract. Please look into this issue and consult native speaker for assistance.
Comments on the Quality of English LanguageThere are several mistakes, for example, in last line of the abstract. Please look into this issue and consult native speaker for assistance.
Author Response
Comment 1:However, when you say several studies, you should add references as well to support those claims. This issue was prominent in Discussion section (Paragraph 4).
Response:
Thank you for this comment. We have meticulously gone through the Discussion section and ensured that every mention of "several studies" or similar generalizations is now accompanied by appropriate citations/references to the supporting literature.
Comment 2:English could be improved. There are several mistakes, for example, in last line of the abstract. Please look into this issue and consult native speaker for assistance.
Response:
Thank you for this comment. The manuscript was edited for English language by MDPI and revised according to the editor's suggestions.
Reviewer 3 Report
Comments and Suggestions for Authors
The authors addressed satisfactorely the concerns.
Author Response
Comment : The authors addressed satisfactorely the concerns.
Response:
Thank you so much for your kind words and for acknowledging that the concerns have been addressed satisfactorily. We truly appreciate the time and effort you have invested in reviewing our manuscript. We are glad to hear that our revisions have met the expectations and we believe that the improvements made have enhanced the quality of our work.